# Clinical Outcomes in Dogs Undergoing Cholecystectomy via a Transverse Incision: A Meta−Analysis of 121 Animals Treated between 2011 and 2021

**DOI:** 10.3390/vetsci10060395

**Published:** 2023-06-15

**Authors:** Hyung-Kyu Chae, Ju-Yeon Jeong, Se-Yoon Lee, Hyun-Min Hwang, Kyoung-In Shin, Jung-Hoon Park, Seo-Yeoun Ji, Yeon-Jung Hong

**Affiliations:** 1Department of Veterinary Internal Medicine, Western Referral Animal Medical Center, Seoul 04101, Republic of Korea; 2Laboratory of Veterinary Internal Medicine, College of Veterinary Medicine, Seoul National University, Seoul 08826, Republic of Korea; 3Department of Veterinary Surgery, Western Referral Animal Medical Center, Seoul 04101, Republic of Korea; 4Department of Statistics, Texas A&M University, College Station, TX 77843, USA; 5Department of Veterinary Radiology, Western Referral Animal Medical Center, Seoul 04101, Republic of Korea

**Keywords:** cholecystectomy, icterus, small-breed dogs, transverse incision

## Abstract

**Simple Summary:**

Cholecystectomy in small-breed dogs through a conventional midline incision requires a long surgical and anesthesia time due to the difficulty in securing the surgical field. In this study, we compare the surgical results of 121 dogs that underwent cholecystectomy over a 10-year period through a transverse incision with previously reported surgical results, and discuss the advantages of transverse incision. Veterinarians with more than 5 years of surgical experience at the referral animal hospital who participated in this study found that it was possible to shorten the surgical time compared with that taken with a conventional midline incision by securing a proper operation view. Comparing with previous studies of cholecystectomy in which the operation time was reported, the dogs in this study had similar indications and perioperative mortality rate, but a significant reduction in the operation time was confirmed through statistical analysis. The results of our study may help in the treatment of dogs in emergency situations where long anesthesia is a burden.

**Abstract:**

Although many studies have been conducted on the use of median and transverse incisions in various surgeries in the field of human medicine, related studies in veterinary medicine are lacking. This study aimed to present treatment options for dogs requiring cholecystectomy by reporting the pros and cons of 121 cholecystectomies performed via transverse incision at our hospital over 10 years. In most included cases, nonelective cholecystectomy was performed in an unstable emergency situation. The perioperative mortality rate was 23.14%, which was not significantly different from that of cholecystectomy performed via the conventional midline approach. However, the overall operation time (46.24 ± 6.13 min; range 35–65 min) was shortened by securing an adequate surgical field of view. The transverse incision approach facilitates fast and accurate surgery without increasing the fatality rate in small-breed dogs, in whom securing an adequate surgical field of view is difficult. Thus, transverse incision should be actively considered in dogs undergoing cholecystectomy due to emergency conditions, such as bile leakage or biliary tract obstruction, since prolonged anesthesia can be burdensome. This study may improve cholecystectomy outcomes in small-breed dogs with difficult-to-secure surgical fields.

## 1. Introduction

Prognosis after surgery depends on treatment timing, surgeon’s experience, and the risk of disease [1,2].

Hepatobiliary surgery is indicated for gallbladder mucocele, biliary obstruction caused by gallstones, hepatobiliary tumors, trauma, and rupture and inflammatory diseases of the hepatobiliary tract. Gallbladder mucocele, defined as a mucus-filled distension of the gallbladder, is a common indication for hepatobiliary surgery in dogs; it can be a life-threatening disease depending on the severity of obstruction and the resulting bile acid leakage [3]. The pathogenesis of gallbladder mucocele is not fully understood, but it is thought to be multi-factorial [4]. Abnormal secretion of mucins, the main component of mucus, can cause various gallbladder diseases. Mechanical obstruction of the cystic duct and common bile duct (CBD) due to the hypersecretion of mucus causes cholestasis. In induced cholestasis, increased exposure of the gallbladder epithelium to cytotoxic bile salts results in increased mucus secretion and cystic mucinous hyperplasia of the gallbladder [4,5]. As this process progresses, mucus cannot flow into the bile duct and causes obstruction. Cholecystectomy and flushing of the CBD are performed to treat post-hepatic jaundice due to biliary obstruction resulting from hepatobiliary disease. Flushing the bile duct during surgery helps improve the flow of bile, and is performed using normograde or retrograde methods [4]. In some cases of bile leakage, intraperitoneal flushing and culture of the bile fluid for antibiotic sensitivity testing and subsequent antibiotic treatment are also necessary.

Cholecystectomy can be performed via laparoscopic surgery in addition to conventional laparotomy in humans [6,7,8]. However, the approach in veterinary cholecystectomy is more uniform, owing to various reasons, such as lack of surgical skills and technique knowledge, financial burden of the owner, and the lack of research on various surgical methods. Although a laparoscopic approach has been attempted in the veterinary field with advantages, such as reduced pain and wound infection risk [9,10,11], the midline incision approach for laparotomy is commonly used in the treatment of small-breed dogs.

Several studies have been conducted on the pros and cons of midline and transverse incisions in the field of human medicine [12,13,14], and appropriate approaches were selected according to the surgeon’s preference. An advantage of transverse incision surgery is that it reduces the incidence of incisional hernia from 14.5% to 1.7% during upper abdominal surgery [13]. 

In the field of veterinary medicine, where there are few reports on the surgical outcomes of transverse incision, most cholecystectomies are performed using midline incisions. The length of incision and duration of surgery during ovariohysterectomy via the lateral flank approach were shorter those that of the midline approach [15]. However, no study has evaluated the pros and cons of the transverse incision approach for cholecystectomy in dogs. To increase knowledge regarding surgical approaches, we assessed the surgical outcomes of cholecystectomy via the transverse incision approach in 121 dogs over a period of 10 years (2011–2021). We retrospectively investigated the records of surgical time, perioperative mortality rate, and duration of hospitalization. This study introduces our surgical method as an alternative or additional treatment option in small-breed dogs requiring cholecystectomy due to biliary tract tumors, cholelithiasis, gallbladder mucocele, and extrahepatic biliary tract obstruction (EHBO). Through the sharing of our 10 years of experience, we would like to present the pros and cons of the transverse incision approach for cholecystectomy in small dogs that are difficult to secure a surgical field. In addition, through a meta−analysis with other studies describing the operation time for cholecystectomy, we compared the anesthesia and operation time required for the transverse incisional method with the operation time of the conventional method. 

## 2. Materials and Methods

### 2.1. Inclusion and Exclusion Criteria 

Client-owned dogs that underwent cholecystectomy via the transverse incision approach at the Western Referral Animal Medical Center (WAMC) between 2011 and 2021 were included in this study. Cholecystectomy was performed in cases of gallbladder mucocele, gallstones, and hepatic or biliary tract tumors causing EHBO and gallbladder rupture. The exclusion criteria included a lack of record of the surgical time and cases in which other procedures were performed concurrently with cholecystectomy. In addition, cases in which methods other than anterograde catheterization were used or the flushing method was not clearly described in the surgical record were also excluded. The criteria for elective or nonelective surgery were similar to those used in previous studies [16]. Surgery performed to prevent future complications related to gallbladder mucocele while the patient’s condition was stable was classified as elective surgery.

### 2.2. Surgical Procedures

In all dogs included in this study, the surgical field of view was secured by creating a transverse incision for cholecystectomy (Figure 1a). The direction of incision was based on the xiphoid process and was perpendicular to the direction of the midline incision along the linea alba, which has a relatively small distribution of blood vessels. The surgeon created a bilateral transverse incision in the rectus abdominis muscle below the costal margin using a surgical blade. After making the incision, exposure was achieved using retractors manually held by an assistant surgeon. The hepatobiliary system was visually inspected by the surgeon, and the gallbladder was separated from the liver and excised. The gallbladder was excised from the liver after separating the capsule and parenchyma of the gallbladder to minimize bleeding. After cholecystectomy, patency was confirmed by anterograde catheterization via the cystic duct to the CBD. A feeding tube of appropriate size was used according to the dilatation of the CBD. After confirming bile flow in the CBD by injecting sterile saline from the cystic duct to the CBD, the cut cystic duct was sutured with an absorbable monofilament suture using a continuous suture pattern. The operation was completed by suturing the peritoneum and the rectus fascia using the continuous suture method. Subcutaneous tissue and skin were closed using the simple interrupted suture method. The primary surgeons who performed the surgeries were all veterinarians at WAMC with more than 5 years of surgical experience (Range: 5–25 years).

### 2.3. Evaluation of Clinical Outcomes Associated with the Transverse Incision Approach

To evaluate the clinical outcomes of the transverse incision approach, operation time, hospitalization duration, complications related to surgical techniques, perioperative death rate, and serologic values, including preoperative bilirubin levels, were investigated. Perioperative death was defined as not surviving until discharge. This included dogs that died in the hospital or were euthanized as they were moribund and not expected to survive. Operative time was defined as the duration from the first incision to the completion of skin suturing after gallbladder removal.

### 2.4. Statistical Analysis 

A meta−analysis using a mixed-effects model was conducted to derive the representative value of the operation time for cholecystectomy in dogs based on previous studies [3,17,18,19,20]. The derived representative value was used as the performance goal of a one-sided, one-sample Wilcoxon signed rank test to statistically confirm that the operative time of the transverse incision approach for cholecystectomy was faster and more accurate than those reported in previous studies [21]. The null and alternative hypotheses are that the median operation time of this study was ‘greater or equal’ and ‘less’ than those of the representative value, respectively. A *p*-value < 0.01 indicated statistical significance.

## 3. Results

### 3.1. Patient Characteristics

In total, 121 client-owned dogs that underwent cholecystectomy via the transverse incision approach at the WAMC were included in this study. Table 1 details the characteristics of the dogs. On categorizing dog breeds as small (<10 kg), medium (10 to <25 kg), and large or giant (≥25 kg) by weight [22], most dogs were small-breed dogs (*n* = 112). The mean age at the time of diagnosis was 10.61 ± 3.47 years (range: 1–18 years). The mean body weight was 5.50 ± 3.46 kg (range: 1.5–24.5 kg). There were 47 spayed females (38.84%), 57 castrated males (47.11%), 6 intact males (4.96%), and 11 intact females (9.09%). As for the distribution of breeds, small breeds such as Maltese, Shih Tzu, Miniature Poodle and Schnauzer, and Yorkshire Terrier were the main species. 

### 3.2. Clinical Findings

The most frequent clinical signs were vomiting (53 dogs, 43.80%), anorexia (38, 31.40%), icterus (22, 18.18%), and lethargy (19, 15.72%). Preoperative blood analyses revealed that the total serum bilirubin levels were abnormally high in the presence of biliary tract obstruction or gallbladder rupture. Depending on the biliary flow and liver function, the total bilirubin value ranged from 0.06 to 40.2 mg/dL. All 121 dogs underwent transabdominal ultrasound examination before surgery. The possibility of gallbladder rupture on ultrasound was judged based on steatitis, the incongruity of the gallbladder wall pericholecystic effusion, or cranial abdominal peritonitis [23]. The need for performing cholecystectomy was determined based on the ultrasound findings, clinical symptoms, serologic values, and medical history.

The leakage of bile fluid or rupture of the gallbladder wall was suspected in 64 dogs based on the ultrasound findings; there was no evidence of leakage/rupture on ultrasound in the remaining 57 dogs. However, 14 of the 57 dogs with no clear evidence of gallbladder rupture on ultrasound showed signs of bile acid leakage and gallbladder rupture during surgery. The presence of gallbladder mucocele (*n* = 93) was the main indication for performing cholecystectomy. Among the dogs with gallbladder mucocele, only 6 dogs (6.45%) underwent elective cholecystectomy, while the remaining 87 (93.55%) underwent emergency surgery due to bile leakage or risk of bile leakage caused by obstruction of biliary outflow. The other indications for performing cholecystectomy included gallstones (*n* = 16), hepatobiliary tumors (*n* = 4), and cases of biliary flow stagnation or obstruction due to inflammatory diseases of the hepatobiliary tract and pancreas (*n* = 8). 

Concurrent medical problems were seen in 50 of the 121 dogs, including myxomatous mitral valve degeneration (MMVD) (*n* = 19), hyperadrenocorticism (*n* = 8), hypothyroidism (*n* = 6), diabetes mellitus (*n* = 3), chronic kidney disease (*n* = 6), pulmonary hypertension (*n* = 4), cystolithiasis (*n* = 2), and idiopathic epilepsy (*n* = 2).

### 3.3. Surgical Results

The perioperative death rate in this study was 23.14% (28/121). The causes of death of the 28 dogs who died in the perioperative period were pancreatitis/peritonitis (*n* = 8), noncardiogenic pulmonary edema (*n* = 4), septic peritonitis (*n* = 3), acute renal failure (*n* = 3), cholangitis (intraoperative flushing was performed smoothly but EHBO reoccurred after surgery) (*n* = 3), disseminated intravascular coagulation (*n* = 3), aspiration pneumonia (*n* = 2), hepatic failure (*n* = 1), and unknown causes (*n* = 1). Although there was no case in which intraoperative normograde catheterization failed to resolve an obstruction, three dogs died due to poor postoperative bile flow resulting from severe cholangitis.

Since an adequate view of the gallbladder may be obtained through the use of a transverse incision, the mean operation time in this study was shorter (46.24 ± 6.13 min, range 35–65 min) than the approximate 100 min duration reported in a previous study with a standard midline incision approach [3]. 

### 3.4. Meta−Analysis Results

The representative value of the operation time required for midline approach cholecystectomy determined by analyzing the data of other studies was 117.68 min (95% confidence interval: 84.75, 150.61) (Figure 2). The results of the Wilcoxon signed rank test yielded a *p*-value of <0.01, which indicates that operative time is shortened when the transverse incision approach is used.

### 3.5. Microbiology and Histologic Findings

Bile was collected during surgery and cultured in 24 of the 121 dogs. Among these, bacterial growth was not confirmed in 18 dogs. *Escherichia coli* (2), *Klebsiella pneumoniae* (1), *Streptococcus equinus* (1), *Escherichia coli and Enterococcus gallinarum* (1), and *Escherichia coli and Enterococcus avium* (1) were identified in the cultures of the remaining six dogs.

Intraoperative liver biopsy was performed in 28 dogs to reveal hepatocellular carcinoma (1), metastatic adenocarcinoma (1), biliary carcinoma (1), hepatocellular adenoma (2), vacuolar hepatopathy (8), hepatitis (4), cholangitis (1), cholangiohepatitis (6), and others including necrosis, hemorrhage, and hyperplasia (4).

### 3.6. Complications Related to Surgical Technique and Duration of Hospitalization

Compared with midline incisions, transverse incisions should be considered with caution because of the increased number of blood vessels distributed throughout incised muscle. Unexpected bleeding while creating the incision increases the risk of infection with mucopurulent exudate due to the presence of bacterial growth in the dead space. In our study, unexpected bleeding could be minimized by careful incision using electric hemostat. In one case, the duration of hospitalization was extended to 22 days owing to inflammation at the surgical site. In this analysis, the mean ± SD duration of hospitalization was 5.93 ± 3.97 days (range, 0–26 days).

## 4. Discussion

The study aimed to introduce the advantages of the transverse incision approach for cholecystectomy in small-breed dogs with difficult-to-secure surgical fields by describing the outcomes of the transverse incision approach, which has been used for several years at WAMC. Ventral midline celiotomy is the most commonly used approach for canine extrahepatic biliary tract surgery at most veterinary clinics [16]. On attempting cholecystectomy via the conventional midline incision approach (Figure 1b), we experienced various complications, such as difficulty in irrigation and postoperative bile leakage due to difficulties securing a surgical field of view. Due to the prevalence of small breeds in South Korea [24], a new approach was suggested to ensure an adequate surgical field. Access through the transverse incision approach enables rapid and precise surgery. Since most dogs undergoing surgery at WAMC are small-breed dogs and difficulty is faced in administering anesthesia due to the severity of disease or instability, the transverse incision approach has been used for cholecystectomy in most cases for the last 10 years. We present the outcomes associated with the use of a midline incision approach. 

Although our ability to compare the outcomes of the approaches is limited, as no control group was included in this study, the transverse incision approach had the following advantages: enhanced visualization of the surgical field facilitating CBD catheterization, which minimized iatrogenic gallbladder rupture or bleeding at the surgical site, and reduced surgical and anesthesia time.

The operation time in this study was shorter than that required for cholecystectomy via the midline approach or laparoscopy in previous studies [3,17,18,19,20] (Figure 2). When our surgeons performed cholecystectomy via the midline incision approach, the operation time was similar to that of previous studies [3,17,18,19,20]. Although a comparative study on the incision approach under similar conditions was not conducted at present, it was found that the transverse incision approach significantly shortened the operation time based on the data assessed in the last 10 years and a meta−analysis of comparisons with studies describing operative times for cholecystectomy in dogs. Thus, securing an appropriate field of view and shortening the operation time are important advantages of the transverse incision approach. In addition, when pulling the gallbladder located in a deep area through the midline incision, excessive gallbladder pulling during the cholecystectomy or flushing the bile duct may cause damage to surrounding organs. The transverse incision approach was able to minimize such surrounding tissue damage.

The parameters other than the operation time were similar to those of other studies in which cholecystectomy was performed. A recent study reported that the perioperative mortality rates of elective and non-elective cholecystectomy differ by 6% in dogs undergoing elective cholecystectomy and 23% in dogs undergoing nonelective cholecystectomy [25]. Although most of our cases were nonelective cholecystectomy due to the characteristics of the cases encountered at WAMC, the perioperative mortality rate of 23.14% (28/121) observed in this study was not significantly different from those of previous studies [3,17,18,19,20]. The transverse incision approach was not considered the cause of death in any case in this study. Causes of death included complications, such as pancreatitis and peritonitis, similar to those reported in a previous study of cholecystectomy outcomes [19]. Not many complications related to surgical technique were reported in this study. Hospitalization was prolonged in 1 dog out of 121 for the management of surgical site inflammation due to unexpected bleeding at the incision site. 

Prolonged duration of surgery and general anesthesia increase the likelihood of perioperative complications [26,27,28,29]. In addition, longer anesthesia time makes it difficult to control the patient’s body temperature while the abdomen is open during surgery, and special measures are required for this. Anesthesia and surgery time reduction through transverse incision will minimize complications related to long anesthesia time.

In the field of human medicine, studies on various incision directions for various types of surgery have been conducted [12]. For example, there are various studies on the benefits and risks of transverse incision or midline incision methods for caesarean section [30,31]. Even in same transverse direction, a comparative analysis of the operation time and postoperative pain was conducted in Joel-Cohen incision and Pfannesitiel incision according to the location [32]. Referring to the reports of pros and cons of surgery performed by various approaches, the surgeon can select an approach suitable for individual preference. 

The purpose of our study is to introduce the advantages of transverse incision, which we have experienced over the past 10 years, such as studies on various approaches to surgery in the field of human medicine. Based on our results, we suggest the use of transverse incision in emergency cholecystectomy due to gallbladder rupture or EHBO. Our data indicate that the transverse incision approach is associated with a shorter operative time without the incidence of serious complications related to surgical techniques, making it an appropriate approach for emergency surgeries where a prolonged duration of general anesthesia is considered risky. 

A limitation of this study is that pulmonary function, length of incision, severity and duration of postoperative pain, or the incidence of incisional hernia were not investigated. Considering that the amount of anesthetic required may be reduced depending on the degree of pain generated during surgery [33], quantitative evaluation of the degree of pain according to the incision method is an important factor in determining the superiority of the surgical method. Sufficient incision length in the transverse direction may shorten the operation time by securing an adequate surgical field of view; however, complications, such as delay in recovery and increased pain, may occur. Additional studies should be conducted on objective pain-related factors, such as the severity of pain due to the wide incision and duration of pain medication use. In addition, the fact that normograde catheterization was performed without enterotomy in all cases is considered a limitation. When comparing postoperative complications such as pancreatitis, bile peritonitis, and persistent gastrointestinal signs and operation time according to normograde CBD catheterization and retrograde CBD catheterization during cholecystectomy in dogs with GB mucocele in a recent study [4], normograde CBD catheterization was associated with faster surgery and it was observed to have fewer complications. When cholecystectomy was performed without catheterization because there was no problem with biliary flow in another study [34], the incidence of postoperative pancreatitis was lower and the operation was shorter than the catheterization group. As such, when cholecystectomy is performed, the direction and method of CBD access can affect the outcome of the operation. 

Since the optimal surgical methods for each dog may differ depending on the degree and presence of bile flow obstruction, the fact that only one direction of CBD flushing was performed in this study may be a limitation. If CBD flushing and catheterization were performed in the optimal way to resolve bile flow for each dog, better perioperative motility and lower complications might have resulted. Our transverse incision method may have facilitated access during normograde flushing, but may provide an unfavorable field of surgical view for retrograde CBD flushing via enterotomy. Further studies are needed on the outcome of cholecystectomy with retrograde CBD catheterization or without catheterization after transverse incision. 

Another limitation was the absence of a control group treated via the midline incision approach. Through the meta−analysis conducted in this study, we tried to prove that our surgery method helps reduce anesthesia and surgery time by comparing the reduction in operation time with the existing veterinary studies in which the operation time was described, but there were inevitably several limitations. The limitation is that the distribution of dogs that underwent cholecystectomy included in the meta−analysis and the skill level of the surgeon were not standardized, and the number of studies used in the meta−analysis due to the small number of papers describing the operation time in the field of veterinary surgery.

Nonetheless, the operation time for the midline incision approach reported by the authors and previous cholecystectomy studies suggest that the transverse incision approach enables more rapid surgery, despite the limitations discussed here. Additional prospective studies are needed to confirm the superiority of the transverse incision approach. 

## 5. Conclusions

Data analysis of the surgical cases that were treated at our hospital over a period of 10 years suggests that the transverse incision approach enables fast and accurate surgery in dogs that require emergency cholecystectomy due to biliary tract obstruction and bile leakage. Since the appropriate surgical method is inevitably different depending on the underlying disease, body shape, and biliary secretion state of the dog that needs cholecystectomy, research on various surgical approaches will help clinicians successfully derive the surgery. This study will likely facilitate the treatment of small-breed dogs undergoing cholecystectomy with difficult-to-secure surgical fields. 

## Figures and Tables

**Figure 1 vetsci-10-00395-f001:**
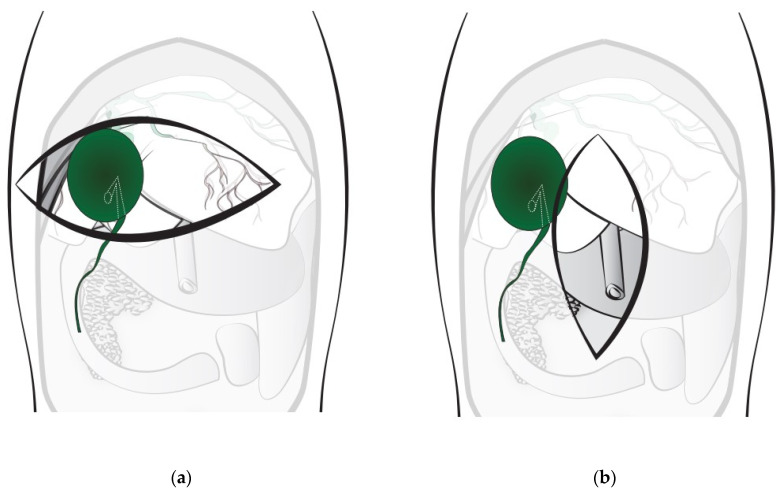
Schematics of the surgical fields obtained via the transverse incision approach (**a**) and conventional midline incision (**b**). Since the gallbladder is located deep on the right side, the surgical field of view associated with the midline incision approach makes cholecystectomy difficult in small-breed dogs.

**Figure 2 vetsci-10-00395-f002:**
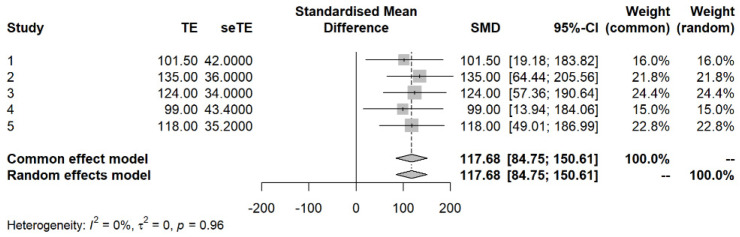
Operation time for cholecystectomy reported by other meta−analyses. The representative value of the time required for an operation was 117.68 min, a value significantly longer than that reported in this study.

**Table 1 vetsci-10-00395-t001:** Signalment of study dogs.

Characteristics	Data (*n* = 121)
Age, median (range)	
Sex	
Castrated males	57
Intact males	6
Spayed females	47
Intact females	11
Weight, median (range)	
<10 kg	112
10 to <25 kg	9
≥25 kg	0
Breeds	
Maltese	25
Shih Tzu	19
Poodle	19
Miniature Schnauzer	14
Yorkshire Terrier	9
Pomeranian	6
American Cocker Spaniel	7
Mixed	5
Miniature Pinscher	4
English Cocker Spaniel	3
Chihuahua	3
Dachshund	1
Siberian Husky	1
Pekinese	1
Japanese Chin	1
Bichon Frise	1
Belgian Malinois	1
Beagle	1

## Data Availability

The data not presented in the manuscript are available for consultation after a reasonable request to the corresponding authors.

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
