# Peer review of "Clinical Outcomes in Dogs Undergoing Cholecystectomy via a Transverse Incision: A Meta−Analysis of 121 Animals Treated between 2011 and 2021"

_vetsci, 2023, doi:10.3390/vetsci10060395_

Round 1
Reviewer 1 Report (Previous Reviewer 1)
I think you have improved the paper and now you are reporting good results using a tenchinque in 100+ dogs at your hospital. That is fine and cannot really lead to strong conclusions for the reasons and limitations you described (no controls, comparisons...etc)
I think there are a few improtant points that need revising. Like including the complications might have been associated to not using enterotomy and papilla flushing...which by any board certified surgeon is a must. It must be mentioned in the discusion as a limitation.
There are a few other comments that need attention. But is very close now.
Regards

Minor edits for grammar and or punctuation are needed.
Author Response
Responses to Reviewer 1
Point 1. I appreciate the developing information about GB mucocele but unless your study is only on surgery for GB mucocele is too much info. Better to reduce to a smaller paragraph.
Response 1. Thank you for your helpful comments. As you advised, we have reduced the scope of our explanation of GB mucocele, as cases other than GB mucocele were included in our study. As shown in the following paragraph, sentences with a strike through were removed: Mucin, a glycoprotein that is a major component of mucus, imparts important viscoelastic properties to bile and contributes significantly to gallbladder mucocele formation. Abnormal secretion of mucins, the main component of mucus, can cause various gallbladder diseases. Mechanical obstruction of the cystic duct and common bile duct (CBD) by the hypersecretion of mucus causes cholestasis. In induced cholestasis, increased exposure of the gallbladder epithelium to cytotoxic bile salts results in increased mucus secretion and cystic mucinous hyperplasia of the gallbladder [4, 5]. As this process progresses, mucus cannot flow into the bile duct and causes obstruction. Predisposing factors for the formation of gallbladder mucocele include endocrinopathy (hypothyroidism, diabetes mellitus, hyperadrenocorticism), hyperlipidemia, and middle to old age. Gallbladder mucoceles are more prevalent in small- and middle-sized dogs, such as Shetland Sheepdogs, Miniature Schnauzers, and Cocker Spaniels [5]. Cholecystectomy and flushing of the CBD are performed to treat post-hepatic jaundice due to biliary obstruction resulting from hepatobiliary disease. Flushing the bile duct during surgery helps improve the flow of bile, and is performed using normograde or retrograde methods [4]. In some cases of bile leakage, intraperitoneal flushing and culture of the bile fluid for antibiotic sensitivity testing and subsequent antibiotic treatment are also necessary.
Point 2. Punctuation, end of brackets and full stop.
Response 2. Thank you for your helpful comments. As per your advice, sentences were revised as follows: Cholecystectomy can be performed via laparoscopic surgery in addition to con-ventional laparotomy in humans [6-8].
Point 3. Repeated sentence unnecessary
Response 3. Thank you for your comment. Based on your advice, the following redundant sentence was removed: Based on these findings, appropriate approaches were selected according to the surgeon’s preference.
Point 4. Is not preferred. It is just a small advantage of this incision that will not cause unmilical hernia.
Response 4. Thank you for the valuable insights. As per your advice, the sentence was revised, as follows: An advantage of transverse incision surgery is that it reduces the incidence of incisional hernia from 14.5% to 1.7% during upper abdominal surgery [13].
Point 5. See previous comment about long paragraph explaining GB mucocele. It was not necessary and excessive as you operated on other conditions that you didn’t explain like you did with GB mucocele.
Point 6. OK, now I get why the extended explanation. I still think you should reduce it in the intro if you are suggesting using this technique for other causes of GB disease
Response 5 & 6. Thank you for the comment. Although GB mucocele was a common indication for surgery in our study, since cholecystectomy was also performed in other cases, sentences related to GB mucocele were deleted or significantly reduced in size in the introduction. As shown in the following paragraph, sentences with a strike through were removed: Mucin, a glycoprotein that is a major component of mucus, imparts important viscoelastic properties to bile and contributes significantly to gallbladder mucocele formation. Abnormal secretion of mucins, the main component of mucus, can cause various gallbladder diseases. Mechanical obstruction of the cystic duct and common bile duct (CBD) by the hypersecretion of mucus causes cholestasis. In induced cholestasis, increased exposure of the gallbladder epithelium to cytotoxic bile salts results in increased mucus secretion and cystic mucinous hyperplasia of the gallbladder [4, 5]. As this process progresses, mucus cannot flow into the bile duct and causes obstruction. Predisposing factors for the formation of gallbladder mucocele include endocrinopathy (hypothyroidism, diabetes mellitus, hyperadrenocorticism), hyperlipidemia, and middle to old age. Gallbladder mucoceles are more prevalent in small- and middle-sized dogs, such as Shetland Sheepdogs, Miniature Schnauzers, and Cocker Spaniels [5]. Cholecystectomy and flushing of the CBD are performed to treat post-hepatic jaundice due to biliary obstruction resulting from hepatobiliary disease. Flushing the bile duct during surgery helps improve the flow of bile, and is performed using normograde or retrograde methods [4]. In some cases of bile leakage, intraperitoneal flushing and culture of the bile fluid for antibiotic sensitivity testing and subsequent antibiotic treatment are also necessary.
Point 7. Dilation of the CBD does not cause bile leakage…needs to change this sentence. Dilation only indicates likely obstruction. You can only say risk of leakage caused by obstruction of biliary outflow…not even just the CBD as could be in the duodenal papilla too
Response 7. Thank you for the valuable insights. As per your advice, the sentence the sentence revised, as follows: Among the dogs with gallbladder mucocele, only 6 dogs (6.45%) underwent elective cholecystectomy, while the remaining 87 (93.55%) underwent emergency surgery due to bile leakage or risk of bile leakage caused by obstruction of biliary outflow.
Point 8. Not sure what you mean. But one does not do biliary flushing but normograde or retrograde biliary CBD flushing….correct sentence
Response 8. Thank you for your valuable comments. Our intent was to describe cases of inflammation of the hepatobiliary tract and pancreas in which surgical attempts were made to resolve obstruction or stasis of bile duct flow that had not been medically resolved. To prevent misunderstanding, the sentence has been modified as follows: The other indications for performing cholecystectomy included gallstones (n=16), hepatobiliary tumors (n=4), and cases of biliary flow stagnation or obstruction due to inflammatory diseases of the hepatobiliary tract and pancreas (n=8).
Point 9. You replied to me about these complications. Really you should write later in the discussion that not performing enterotomy to flush from papilla(gold standard) might have been the cause of this.
Response 9. Thank you for the comment. As you advised, we mentioned in the discussion that enterotomy, the gold standard treatment, was not performed for flushing and may be a limitation. In the text, the following was added: In addition, the fact that normograde catheterization was performed without enterotomy in all cases is considered a limitation. Although this method is thought to have made rapid surgery possible, it may have caused 3 cases of unresolved postoperative EHBO. Further research on the flushing method after transverse incision is needed.
In addition, since unresolved EHBO may be misinterpreted, the text was altered as follows: he causes of death of the 28 dogs who died in the perioperative period were pancreatitis/peritonitis (n=8), noncardiogenic pulmonary edema (n=4), septic peritonitis (n=3), acute renal failure (n=3), cholangitis (intraoperative flushing was performed smoothly but EHBO reoccurred after surgery) (n=3), disseminated intravascular coagulation (n=3), aspiration pneumonia (n=2), hepatic failure (n=1), and unknown causes (n=1).
Point 10. Again, same as previous comment. Any surgeon I know would right away reject this paper based on this. So the only way this paper could be accepted by any board certified surgeon is to acknowledge this was not done and that could have been the cause of complications.
Response 10. Thank you very much for pointing out an important point. Our intention was that there were no cases in which the obstruction was not resolved smoothly when attempting intraoperative normograde cathetherization, but there were 3 cases of death due to poor postoperative bile flow due to severe cholangitis. To avoid confusion, the sentence was revised once again, and the limitations were specified in the discussion. Revised sentences were as follows: Although there was no case in which intraoperative normograde catheterization failed to resolve an obstruction, 3 dogs died due to poor postoperative bile flow resulting from severe cholangitis.
Point 11. Back to the introduction description of GB mucocele. There are many other pathologies you found but were not described, that is why I would not expand so much in the intro as this paper does not only suggest the transverse approach for GB mucocele
Response 11. Thank you for your comments. As mentioned above, many sentences in the introduction that focused on the pathogenesis of GB mucocele have been revised. Original sentences with a strikethrough were removed from the text after revision, as shown in the following paragraph: Mucin, a glycoprotein that is a major component of mucus, imparts important viscoelastic properties to bile and contributes significantly to gallbladder mucocele formation. Abnormal secretion of mucins, the main component of mucus, can cause various gallbladder diseases. Mechanical obstruction of the cystic duct and common bile duct (CBD) by the hypersecretion of mucus causes cholestasis. In induced cholestasis, increased exposure of the gallbladder epithelium to cytotoxic bile salts results in increased mucus secretion and cystic mucinous hyperplasia of the gallbladder [4, 5]. As this process progresses, mucus cannot flow into the bile duct and causes obstruction. Predisposing factors for the formation of gallbladder mucocele include endocrinopathy (hypothyroidism, diabetes mellitus, hyperadrenocorticism), hyperlipidemia, and middle to old age. Gallbladder mucoceles are more prevalent in small- and middle-sized dogs, such as Shetland Sheepdogs, Miniature Schnauzers, and Cocker Spaniels [5]. Cholecystectomy and flushing of the CBD are performed to treat post-hepatic jaundice due to biliary obstruction resulting from hepatobiliary disease. Flushing the bile duct during surgery helps improve the flow of bile, and is performed using normograde or retrograde methods [4]. In some cases of bile leakage, intraperitoneal flushing and culture of the bile fluid for antibiotic sensitivity testing and subsequent antibiotic treatment are also necessary.
Point 12. which consists in severing muscles at aponeurosis/less thickness of muscle/less vascular…midline incissions are not less vascular….muscles are less vascular…
Response 12. Thank you for the comment. To avoid confusion, the sentence has been modified as follows: Compared with midline incisions, transverse incisions should be considered with caution because of the increased number of blood vessels distributed throughout incised muscle.
Point 13. You cannot say the midline approach is poor surgical technique. If you mean the surgeons poor surgical technique then is admitting the surgeons were not great. Maybe say to lack of visibility associated with midline approach…but do not judge that technique as it is the standard technique and is your word against so many other surgeons including specialists using it…
Response 13. Thank you very much for mentioning this very important point. The conclusion of this study is not the superiority of our surgical method but instead its introduction. In small-breed dogs, an advantage of the method is shortening the surgical time by securing the surgical field. To clarify our meaning, the sentence was amended, as follows: On attempting cholecystectomy via the conventional midline incision approach (Figure 1b), we experienced various complications, such as difficulty in irrigation and postoperative bile leakage due to difficulties securing a surgical field of view.
Point 14. Again….was not needed….but you could say was suggested or favourable in the author’s opinion
Response 14. Thank you for the comment. The sentence has been modified, as follows: Due to the prevalence of small breeds in South Korea [26], a new approach was suggested to ensure an adequate surgical field.
Point 15. Needs rephrasing…grammar sounds wrong. …has been used for cholecystectomy in most cases for the last 10 year…
Response 15. Thank you for the comment. The sentence has been modified, as follows: Since most dogs undergoing surgery at WAMC are small-breed dogs and difficulty is faced in administering anesthesia due to the severity of disease or instability, the transverse incision approach has been used for cholecystectomy in most cases for the last 10 years
Point 16. It is very subjective….you cannot compare so I would delete Fewer…but you can say that.. Not many complications were reported….etc
Response 16. Thank you for the comment. The sentence, which can be subjective, has been modified, as per your advice: Not many complications related to surgical technique were reported in this study.
Point 17. Strong statement. Maybe better We suggest/we support/we endorse
Response 17. Thank you for the comment. The sentence has been modified as follows: Based on our results, we suggest the use of transverse incision in emergency cholecystectomy due to gallbladder rupture or EHBO.
Point 19. More rapid…..I think this sentence you are comparing methods so you would need to use a comparative adjective
Response 19. Thank you for the comment. The sentence has been modified as follows: Nonetheless, the operation time for the midline incision approach reported by the authors and previous cholecystectomy studies suggest that the transverse incision approach enables more rapid surgery, despite the limitations discussed here.

Reviewer 2 Report (Previous Reviewer 2)
I thank you for the revision work you have done. The work behind the revision of the paper is satisfactory in my opinion.
Author Response
Responses to Reviewer 2.
I thank you for the revision work you have done. The work behind the revision of the paper is satisfactory in my opinion.
Response. Thank you very much for reviewing our study. We look forward to moving this manuscript closer to being ready for publication in Veterinary Sciences.

Reviewer 3 Report (New Reviewer)
I read this article with great interest. Authors presented very practical information in cholecystectomy survey in dogs. In my opinion each soft tissue surgeon should read this manuscript. All chapters are planned and written very properly. Statistics analysis is very helpful and good. I am sure that this article has to be published in Veterinary Sciences.
Author Response
Responses to Reviewer 3.
I read this article with great interest. Authors presented very practical information in cholecystectomy survey in dogs. In my opinion each soft tissue surgeon should read this manuscript. All chapters are planned and written very properly. Statistics analysis is very helpful and good. I am sure that this article has to be published in Veterinary Sciences.
Response. Thank you very much for reviewing our manuscript. We look forward to moving this manuscript closer to being publication ready.

Reviewer 4 Report (New Reviewer)
Dear Author, Thank you for the opportunity to review this manuscript.
The meta-analysis work presented certainly provides information that enriches the present bibliography. Congratulations on evaluating the potency of the effect of the operative technique.
It would be interesting to provide some observations on the anesthetic technique and on the request for intra- and post-operative analgesic. Describe the techniques by which anesthetists established the cut-of point for the delivery of rescue analgesia.
I enclose the following bibliographical notes from which you can draw inspiration to improve your interesting metanalysis.
Giovanna L Costa, Simona Di Pietro, Claudia Interlandi*, Fabio Leonardi, Daniele Macrì, Vincenzo Ferrantelli, Francesco Macrì. Effect on physiological parameters and anaesthetic dose requirement of isoflurane when tramadol given as a continuous rate infusion vs a single intravenous bolus injection during ovariohysterectomy in dogs. PLoS ONE 18(2), e0281602
Minor editing of English language required
Author Response
Responses to Reviewer 4.
Dear Author, Thank you for the opportunity to review this manuscript. The meta-analysis work presented certainly provides information that enriches the present bibliography. Congratulations on evaluating the potency of the effect of the operative technique. It would be interesting to provide some observations on the anesthetic technique and on the request for intra- and post-operative analgesic. Describe the techniques by which anesthetists established the cut-of point for the delivery of rescue analgesia. I enclose the following bibliographical notes from which you can draw inspiration to improve your interesting metanalysis.
Giovanna L Costa, Simona Di Pietro, Claudia Interlandi*, Fabio Leonardi, Daniele Macrì, Vincenzo Ferrantelli, Francesco Macrì. Effect on physiological parameters and anaesthetic dose requirement of isoflurane when tramadol given as a continuous rate infusion vs a single intravenous bolus injection during ovariohysterectomy in dogs. PLoS ONE 18(2), e0281602
Response. Thank you very much for reviewing our study. We will refer to the reference you gave me when we do additional research. We look forward to moving this manuscript closer to being ready for publication in Veterinary Sciences.

Round 2
Reviewer 1 Report (Previous Reviewer 1)
My previous comments have been addressed
This manuscript is a resubmission of an earlier submission. The following is a list of the peer review reports and author responses from that submission.
Round 1
Reviewer 1 Report
Dear authors. I agree with some of the conclusion but you are being categorical when you present your data. I think you can only suggest this approach might have overall benefits but not necessarily for every patient. Also you have not presented difference or data compared to midline incision, specially in your hospital. You mostly do transversal incisions hence you are less prone to do midline. You would need to compare both in similar settings (not just studies, general studies), but similar setting, like similar skill from the surgeons, similar morbidity at presentation, a referral case might be already very sick and survival will not be great. Your mortality rate is in the end of the range of mortality you reported, hence, is difficult to convince the reader the transversal vs the midline approach is better for the animal. I think this study has merit and can be valuable but the way you present it is arguable at the least. You cannot prove some of the statements you say. Please refer to the comments in the document for more details.

Author Response
February 16, 2023
Prof. Dr. Patrick Butaye
Editor-in-Chief
Veterinary Science
Dear Dr. Butaye:
We wish to re-submit the manuscript titled “Clinical outcomes in dogs undergoing cholecystectomy via a transverse incision: A meta-analysis of 121 animals treated between 2011 and 2021”. The manuscript ID is vetsci-2201372.
We thank you and the reviewers for your thoughtful suggestions and insights. The manuscript has benefited from these insightful suggestions. I look forward to working with you and the reviewers to move this manuscript closer to publication in Veterinary Sciences.
The manuscript has been rechecked and the necessary changes have been made in accordance with the reviewer’s suggestions. The revisions in the manuscript are indicated by the red font and the responses to all comments have been prepared and given below.
Thank you for your consideration. We look forward to hearing from you.
Sincerely,
Yeon-Jung Hong, D.V.M., MS
Chief Veterinarian
Department of Veterinary Surgery, Western Referral Animal Medical Center, Seoul 04101, Republic of Korea
E-mail: vethong@hanmail.net
Responses to Reviewer 1
Thank you for your kind comment.
Based on your advice, We have corrected the sentences and marked them in red.
Please see attached file.

Reviewer 2 Report
the interest and originality of the article is remarkable.
The reading also flows pleasantly. In my opinion, no substantial revisions of the work are necessary. The surgical procedure on the type of incision should be better described. The results section should be revised in form.

Author Response
February 16, 2023
Prof. Dr. Patrick Butaye
Editor-in-Chief
Veterinary Science
Dear Dr. Butaye:
We wish to re-submit the manuscript titled “Clinical outcomes in dogs undergoing cholecystectomy via a transverse incision: A meta-analysis of 121 animals treated between 2011 and 2021”. The manuscript ID is vetsci-2201372.
We thank you and the reviewers for your thoughtful suggestions and insights. The manuscript has benefited from these insightful suggestions. I look forward to working with you and the reviewers to move this manuscript closer to publication in Veterinary Sciences.
The manuscript has been rechecked and the necessary changes have been made in accordance with the reviewer’s suggestions. The revisions in the manuscript are indicated by the red font and the responses to all comments have been prepared and given below.
Thank you for your consideration. We look forward to hearing from you.
Sincerely,
Yeon-Jung Hong, D.V.M., MS
Chief Veterinarian
Department of Veterinary Surgery, Western Referral Animal Medical Center, Seoul 04101, Republic of Korea
E-mail: vethong@hanmail.net
Responses to Reviewer 2
Point 1. Line 21: is it possible to better understand what is meant by well-trained? this is something that should be explored further perhaps in the materials and methods section. For example, did the veterinary surgeons who performed the surgery have the same years of surgical experience behind them? What is the minimum number of cholecystectomies with both surgical approaches considered necessary to be well-trained?
Response 1. Thank you for your helpful comments. Well-trained used to mean that a person has sufficient experience by working for more than 5 years in the surgical field of our referral hospital. The following has been modified to prevent confusion; Veterinarians with more than 5 years of surgical experience at the referral animal hospital who participated in this study found that it was possible to shorten the surgical time compared with that taken with a conventional midline incision by securing a proper operation view.
Point 2. Introduction
Line 71: The introduction is fluent and understandable. Several techniques have been described in human medicine with relative outcomes for both the midline and transverse approaches. In veterinary medicine, on the other hand, laparotomic approaches via midline and laparoscopy have been described. It is not clear whether the transverse approach has actually been described or not, can this be clarified? The article does indeed talk about clinical outcomes of this surgical technique, but it is not clear whether this technique has been described in detail yet.
Response 2. Thank you for your helpful comments. To our knowledge, there are not many studies on the difference in surgical results according to the incision approach in the veterinary field. Only comparative reports of lateral frank incision and midline incision in ovariohysterectomy have been reported. To clarify the scarcity of our research, we have added the following paragraph as you pointed out; In the veterinary field, where there are few reports on the surgical outcomes of transverse incision, most cholecystectomies are performed using midline incisions. There are only reports that the length of incision and duration of surgery are shortened during ovariohysterectomy in the lateral flank approach other than the midline ap-proach. In case of cholecystectomy in dogs, no study introduced the pros and cons of the transverse incision approach. To increase knowledge related to surgical approaches, we assessed surgical outcomes of cholecystectomy via a transverse incision in 121 dogs over a period of 10 years (2011–2021). Based on our experience of securing an appropriate surgical field, we performed cholecystectomy through a transverse incision in a number of dogs and retrospectively investigated the records of surgical time, perioperative mortality rate, duration of hospitalization, etc.
Point 3. Materials and methods
Surgical procedures
The aim of the meta-analysis is to identify the outcome of the transverse incision, but this first step is described in a hasty manner. The surgical procedure should be described in more detail precisely on the type of incision. Identify the repere points, identify tissues and structures involved during surgical access and how these are cut.
Response 3. Thank you for your comment. As your advice, related sentence was modified as follows: In all dogs included in this study, the surgical field of view was secured by performing a transverse incision for cholecystectomy (Figure 1a). First, the surgeon performed a bilateral transverse incision of the rectus abdominis muscle below the costal margin using a surgical blade on all dogs. After making the incision, exposure was achieved using retractors manually held by an assistant surgeon. Then, hepatobiliary system was visually inspected by the surgeon. The gallbladder was separated from the liver and excised. In this process, the gallbladder was excised from the liver after separating the capsule and parenchyma of the gallbladder to minimize the occurrence of bleeding. After cholecystectomy, patency was confirmed by anterograde catheterization via the cystic duct to CBD. A feeding tube of appropriate size was used according to the dilatation of the CBD. After confirming the bile flow in the CBD by injecting sterile saline from the cystic duct to the CBD, the cut cystic duct was sutured using an absorbable monofilament suture using a continuous suture method. After this, the operation was completed by suturing the peritoneum and the rectus fascia using a continuous suture method. Subcutaneous tissue and skin were closed using simple interrupted suture method.
Point 4. Materials and methods
Evaluation of clinical outcomes associated with the transverse incision approach
106: “adverse events” is a general definition, what does it refer to? problems related to surgical technique/management such as bleeding or suture dehiscence for example? or problems related to the patient's general condition? E.g. possible anorexia/dysorexia, inadequate pain management etc. In fact, if you read on the 'clinical findings' section you will read a number of reported 'adverse events'. So it would be interesting to classify them precisely so that it would be more structured and orderly.
Response 4. Thank you for the valuable insights. To prevent confusion, “adverse events” has been changed to “problem related to surgical technique”. Related sentence was modified as follows: To evaluate clinical outcomes of the transverse incision approach, operation time, hospitalization duration, problem related to surgical techniques, perioperative death rate, and serologic values including preoperative bilirubin levels were investigated.
Point 5. Results
Surgical results
Only objective data should be entered and presented in the results section. Considerations and discussions should be placed in the dedicated section. The whole section is a series of data already discussed. In my opinion, it should be reworked so as to reduce the discussions in this section. Ex. Line: 158-164 is discussion, do not result.
Response 5. We apologize for the confusion. As you advised, sentences other than objective data have been moved to discussion.
Point 6. Adverse events and duration of hospitalization
Line 192: “Adverse events” Consistent with the above, adverse events should be classified and therefore this section should also be readjusted.
Response 6. Thank you for pointing this out. We have revised the sentence as follows: 3.6. Problem related to surgical technique and duration of hospitalization

Reviewer 3 Report
In my opinion this article is not very interesting and it is badly designed. Reading the title gives you indications that it is a review of the surgical technique itself and its comparison with ventral midline celiotomy. However, the study variables do not indicate anything about the pain and inflammation that this technique can cause, which seems essential to me. Nor is it included which anesthetic and/or analgesic protocols have been used.
For all these general reasons about the article, I am very sorry to have to decline its acceptance for publication.
Author Response
February 16, 2023
Prof. Dr. Patrick Butaye
Editor-in-Chief
Veterinary Science
Dear Dr. Butaye:
We wish to re-submit the manuscript titled “Clinical outcomes in dogs undergoing cholecystectomy via a transverse incision: A meta-analysis of 121 animals treated between 2011 and 2021”. The manuscript ID is vetsci-2201372.
We thank you and the reviewers for your thoughtful suggestions and insights. The manuscript has benefited from these insightful suggestions. I look forward to working with you and the reviewers to move this manuscript closer to publication in Veterinary Sciences.
The manuscript has been rechecked and the necessary changes have been made in accordance with the reviewer’s suggestions. The revisions in the manuscript are indicated by the red font and the responses to all comments have been prepared and given below.
Thank you for your consideration. We look forward to hearing from you.
Sincerely,
Yeon-Jung Hong, D.V.M., MS
Chief Veterinarian
Department of Veterinary Surgery, Western Referral Animal Medical Center, Seoul 04101, Republic of Korea
E-mail: vethong@hanmail.net
Responses to Reviewer 3
Thank you for reviewing our study. We will take your points into consideration and try to reflect them in the next study.

Round 2
Reviewer 1 Report
Dear authors. Thank you for resubmitting the manuscript. I think it has improved.
I believe your message is clear and what you are trying to deliver is an alternative or additional approach to cholecystectomy. However, you are repeating and duplicating or triplicating your message in areas like in the discussion. I don't think you can say is better than others as you don't have the criteria to decide how is better or worse. You can only report and describe what you did and your findings. You don't have controls, comparison, pain scores not done to assess whether transverse approach causes more pain, the mortality rate is the same, stays in the high end of the non elective surgeries. That means is not decreasing mortality. What about comparison of hospitalisation times too? Other markers for pain or for response...? You only mention 1 complication and that is great but are there not other cons to this approach? Bigger muscle disection or severing larger bellies causing more bleeding and pain? I think your message is valid and clear and is publishable as you have a good number of cases. Just the way is presented to the reader appears to compare like being better. You have not proven is better and is not the scope of your study. You can certainly say is that reduces anesthesia time (sure) and that improves visibility. Some more limitations would need to be addressed in the discussion. Overall is better. But the message needs to be more clear and objective. This is an additional or alternative (not necessarily better until you can prove it) treatment. Very acceptable to do in cases that warrant it. Another advise is to soften the welfare comments as if you believe is a welfare issue to use the tranverse approach we would have no option but reject this project. Welfare is top above everything in veterinary medicine. I made some suggestion for edits and comments in the manuscript. Please check and resubmit when all is addressed.

Author Response
March 14, 2023
Prof. Dr. Patrick Butaye
Editor-in-Chief
Veterinary Science
Dear Dr. Butaye:
We wish to re-submit the manuscript titled “Clinical outcomes in dogs undergoing cholecystectomy via a transverse incision: A meta-analysis of 121 animals treated between 2011 and 2021”. The manuscript ID is vetsci-2201372.
We thank you and the reviewers for your thoughtful suggestions and insights. The manuscript has benefited from these insightful suggestions. I look forward to working with you and the reviewers to move this manuscript closer to publication in Veterinary Sciences.
The manuscript has been rechecked and the necessary changes have been made in accordance with the reviewer’s suggestions. The revisions in the manuscript are indicated by the red font and the responses to all comments have been prepared and given below.
Thank you for your consideration. We look forward to hearing from you.
Sincerely,
Yeon-Jung Hong, D.V.M., MS
Chief Veterinarian
Department of Veterinary Surgery, Western Referral Animal Medical Center, Seoul 04101, Republic of Korea
E-mail: vethong@hanmail.net
Dear authors. Thank you for resubmitting the manuscript. I think it has improved.
I believe your message is clear and what you are trying to deliver is an alternative or additional approach to cholecystectomy. However, you are repeating and duplicating or triplicating your message in areas like in the discussion. I don't think you can say is better than others as you don't have the criteria to decide how is better or worse. You can only report and describe what you did and your findings. You don't have controls, comparison, pain scores not done to assess whether transverse approach causes more pain, the mortality rate is the same, stays in the high end of the non elective surgeries. That means is not decreasing mortality. What about comparison of hospitalisation times too? Other markers for pain or for response...? You only mention 1 complication and that is great but are there not other cons to this approach? Bigger muscle disection or severing larger bellies causing more bleeding and pain? I think your message is valid and clear and is publishable as you have a good number of cases. Just the way is presented to the reader appears to compare like being better. You have not proven is better and is not the scope of your study. You can certainly say is that reduces anesthesia time (sure) and that improves visibility. Some more limitations would need to be addressed in the discussion. Overall is better. But the message needs to be more clear and objective. This is an additional or alternative (not necessarily better until you can prove it) treatment. Very acceptable to do in cases that warrant it. Another advise is to soften the welfare comments as if you believe is a welfare issue to use the tranverse approach we would have no option but reject this project. Welfare is top above everything in veterinary medicine. I made some suggestion for edits and comments in the manuscript. Please check and resubmit when all is addressed.
Point 1. Amongst what? among the causes, the inciting causes? does not make sense otherwise. This sentence needs to start differently. It does not make sense otherwise. Amongst what? amongnst causes? elements? The reader will not know otherwise what you mean
Response 1. Thanks for pointing out some confusing sentences. To prevent confusion, we modified the sentence that starts with among and added an explanatory sentence. The revised sentences were as follows: Mucin, a glycoprotein that is a major component of mucus, is thought to impart important viscoelastic properties to bile and contribute significantly to gallbladder mucocele formation. Abnormal secretion of these mucins can cause various gallbladder disease. Mechanical obstruction of the cystic duct and common bile duct (CBD) by hypersecreted mucus causes cholestasis. In the state of induced cholestasis, increased exposure of cytotoxic bile salts in the gallbladder epithelium results in in-creased mucus secretion and sometimes cystic mucinous hyperplasia of the gallbladder [4]. As this process accumulates, mucus cannot escape into the bile duct and causes obstruction. (Line56-64)
Point 2. This sentence is unnecessary and if kept better rephrased differently.
Response 2. Thanks for pointing out an important point. We have removed the seemingly unnecessary sentences according to your advice.
Point 3. In your opinion securing a surgical fields is key to success….maybe that would read better. I think is not necessary and this would be best part of discussion. This is the introduction. This sentence is confusing, does not add and is bias. You want to say that you prioritize surgical field or consider an appropriate surgical field a key for success. Rephase or delete the sentence. This is the introduction part anyway.
Response 3. Thanks for pointing out an important point. The sentence has been modified so that the opinions of the authors are not reflected in the introduction.
Point 4. In your opinion on in the authors opinion the results of this study suggest….potential improvement.
Response 4. Thanks for the advice. Based on your advice, the significance of this study has been modified as follows. This study introduces our surgical method as an alternative or additional treatment option for small dog breeds requiring cholecystectomy due to procedures such as biliary tract tumors, cholelithiasis, gallbladder mucocele, and extrahepatic biliary tract obstruction (EHBO).
Point 5. Did you not have any problem with obstructions that you couldn’t catheterise? That’s the reason of enterotomies to access via the papilla instead. Understand that is the gold standard. maybe need to be at least mentioned in the discussion. My previous comment about if some dogs might have not been unblocked. Did all have appropriate flushing? Without doing enterotomy?
Response 5. Thanks for pointing out an important point. Fortunately, there were no cases in this study that required catheterization via enterotomies because the obstruction did not resolve intraoperatively. However, there were cases where jaundice did not improve and eventually died due to poor bile flow due to severe pancreatitis or cholecystitis after surgery, so a related sentence was added as follows: There was no case where intraoperative anterograde catheterization attempts did not resolve the obstruction, but there were 11 cases that resulted in death due to poor postoperative bile flow due to severe cholangitis or pancreatitis.
Point 6. What about more pain due to the more muscle being trabnversed through belly? Was that a problem you encountered?
Response 6. Thanks for pointing out an important point. Unfortunately, the investigation of objective indicators related to pain was not included in the study, so the following sentence was added to the limitations of the study: Additional studies on objective pain-related factors such as the severity of pain due to the wide incision and duration of pain medication use will be needed.
Point 7. It is the same advantage. You would need to rephrase this. Improves the surgical field and facilitates CBD catheterisation due to that improved visualisation….therefore minimising the its iatrogenic rupture and shortening as well the surgical time.
Response 7. Thanks for the advice. Based on your advice, the sentence has modified as follows: Enhanced surgical field of vision facilitates CBD catheterization and this minimizes iatrogenic gallbladder rupture or bleeding at the surgical site and shortens the surgical and anesthesia time.
Point 8. The results are the same. I would rephrase to say that mortality is the same with transverse approach but surgical time is reduced…visualisation and catheterisation of the CBD improved and overall success not in ferior… hence the advantage of its use according to your results.
All of this is repeated above
Response 8. Thanks for pointing out an important point. We have removed sentences that appear to be repeated.
Point 9. Important limitations that need to be considered (everything is animal wefare and if you focus on being a welfare issue you would not be allowed to do it in any country that protects welfare. So I sggest rephrase and soften to something similar to what I wrote above.
Response 9. Thanks for pointing out an important point. The sentence related to animal welfare was deleted, and the sentence that research including objective factors related to pain should be performed was added.

Reviewer 3 Report
As I already commented in the other review, and despite the fact that the review carried out by the authors has improved substantially, it does not seem to me that it addresses the most important issues that should be taken into account when comparing both types of techniques. It does not seem to me that they compare variables that provide us with relevant information to verify which technique is better. For all this, I have to reject your publication on my part.
Author Response
March 14, 2023
Prof. Dr. Patrick Butaye
Editor-in-Chief
Veterinary Science
Dear Dr. Butaye:
We wish to re-submit the manuscript titled “Clinical outcomes in dogs undergoing cholecystectomy via a transverse incision: A meta-analysis of 121 animals treated between 2011 and 2021”. The manuscript ID is vetsci-2201372.
We thank you and the reviewers for your thoughtful suggestions and insights. The manuscript has benefited from these insightful suggestions. I look forward to working with you and the reviewers to move this manuscript closer to publication in Veterinary Sciences.
The manuscript has been rechecked and the necessary changes have been made in accordance with the reviewer’s suggestions. The revisions in the manuscript are indicated by the red font and the responses to all comments have been prepared and given below.
Thank you for your consideration. We look forward to hearing from you.
Sincerely,
Yeon-Jung Hong, D.V.M., MS
Chief Veterinarian
Department of Veterinary Surgery, Western Referral Animal Medical Center, Seoul 04101, Republic of Korea
E-mail: vethong@hanmail.net
As I already commented in the other review, and despite the fact that the review carried out by the authors has improved substantially, it does not seem to me that it addresses the most important issues that should be taken into account when comparing both types of techniques. It does not seem to me that they compare variables that provide us with relevant information to verify which technique is better. For all this, I have to reject your publication on my part.
Response. Thank you for reviewing our study. As advised by you, the manuscript has been significantly revised for the purpose of introducing the advantages of our method over the past 10 years due to the lack of control group and factors to prove that this incision method is superior. We will try to carry out additional research by reflecting the lacking part. Thank you for reviewing the study.
